# Hyperbolic Embedding Inference for Structured Multi-Label Prediction

**Bo Xiong**[*]
University of Stuttgart
Stuttgart, Germany

**Michael Cochez**
Vrije Universiteit Amsterdam
Discovery Lab, Elsevier
Amsterdam, The Netherlands

**Mojtaba Nayyeri**
University of Stuttgart
Stuttgart, Germany

**Steffen Staab**
University of Stuttgart
University of Southampton
Stuttgart, Germany

## Abstract

We consider a structured multi-label prediction problem where the labels are organized under implication and mutual exclusion constraints. A major concern is to produce predictions that are logically consistent with these constraints. To do so, we formulate this problem as an *embedding inference* problem where the constraints are imposed onto the embeddings of labels by *geometric construction*. Particularly, we consider a hyperbolic Poincaré ball model in which we encode labels as Poincaré hyperplanes that work as linear decision boundaries. The hyperplanes are interpreted as convex regions such that the logical relationships (implication and exclusion) are geometrically encoded using *insideness* and *disjointedness* of these regions, respectively. We show theoretical groundings of the method for preserving logical relationships in the embedding space. Extensive experiments on 12 datasets show 1) significant improvements in mean average precision; 2) lower number of constraint violations; 3) an order of magnitude fewer dimensions than baselines.

## 1 Introduction

Structured multi-label prediction is a task aiming to associate every object with multiple labels that are semantically constrained in a structured manner (e.g., by implication and exclusion constraints). This task is of growing importance in many applications such as image annotation [1, 2], text categorization [3, 4] and functional genomics [5, 6]. One of the central concerns of the task is to produce predictions that are *logically consistent* with the constraints of the labels. For example, a protein must be labeled to have the function *nucleic acid binding* if it is already labeled to have the function *RNA binding* (i.e., implication) and must not have the function *drug binding* (i.e., mutual exclusion).

Various works have been proposed to improve the prediction consistency [7, 8, 9, 10, 11]. One line of work called *label embedding* aims to represent labels as low-dimensional vectors [12, 13]. A key disadvantage of the vector-based representations is that they only capture weak forms of correlation or "similarity' between labels, but do not strongly enforce the logical relationships. Another line of work [7, 9, 14, 8] imposes these logical constraints directly to the losses of neural networks. However, they do not explicitly learn the representations of labels and typically require a complete label taxonomy, which is not always available in and scalable to real-world settings [11].

---

[*]Correspondence to bo.xiong@ipvs.uni-stuttgart.de

36th Conference on Neural Information Processing Systems (NeurIPS 2022).

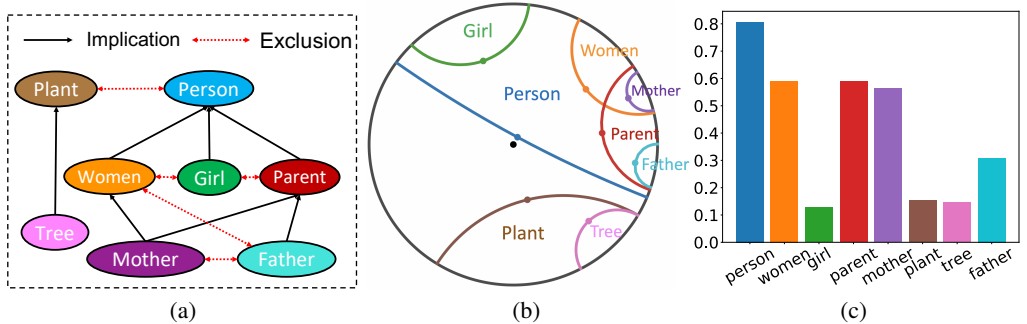

Figure 1: (a) A HEX graph describing the logical relationships (implication and exclusion) between different labels; (b) The learned label embeddings (linear decision boundaries) in the Poincaré ball, where all constraints in the HEX graph are respected; (c) The prediction scores of a given instance of *mother* respect all constraints in the HEX graph, where each score is calculated as the confidence of the instance embedding being a member of the convex region of the corresponding label embedding.

Embedding-based inference [15], which imposes logical constraints directly to the label embeddings, is able to *inductively* infer the underlying label relationships from incomplete labelings [16]. Once all embeddings are adhering to the constraints, each label can be predicted independently without accessing the label relationships, which significantly reduces the computation cost during inference [15]. The key idea, which is inspired by the Venn diagram [16, 17] or set-theoretic semantics [18], is to represent each label as a convex region [15]. A prominent example is the multi-label box model (MBM) [11] that models label implications as box containments. However, MBM learns box-like decision boundaries, which are typically not compatible with standard classifiers (i.e., hyperplane margin-based models such as logistic regression [19]). Besides, box models suffer from a theoretical limitation, i.e., lower-way intersections enforce higher-way interactions [20]. Finally, current methods ignore the importance of constraining mutual exclusion, which is essential as otherwise, a model could trivially obtain zero implication violation by assigning the same score to all labels.

In this paper, we consider a structured multi-label prediction problem with *implication* and *mutual exclusion* constraints that are jointly described by a hierarchy and exclusion (HEX) graph (see Figure 1(a) for an example). The key idea of our method is to transform the logical constraints into soft geometric constraints in the embedding space. In particular, we consider a hyperbolic Poincaré ball model that has demonstrated advantages in representing hierarchical data [21, 22] and assign each label a Poincaré hyperplane that has several favorable theoretical properties in classification. Each Poincaré hyperplane can be interpreted as a convex region such that the *implication* and *mutual exclusion* are modeled by geometric *insideness* and *disjointness* between the corresponding regions, respectively. In this way, a multi-label classifier can be defined by measuring the confidence of an instance having a label as geometric *membership*. Unlike other hyperbolic region-based models such as hyperbolic cones [23] and hyperbolic disks [24], Poincaré hyperplane works as a linear decision boundary and can be seamlessly integrated into existing margin-based classifiers such as hyperbolic logistic regression [19]. Figure 1(b) shows an example of the learned label representations that respect all the constraints given in Figure 1(a). We show theoretical groundings of the proposed method on modeling *implication* and *mutual exclusion*. Extensive experiments on 12 multi-label classification tasks show the model's capability to improve the mean average precision significantly while keeping the number of constraint violations low and requiring an order of magnitude fewer dimensions.

## 2 Preliminaries

**Poincaré ball model** The Poincaré ball $\left(\mathbb{D}^n, g^{\mathbb{D}}\right)$ is one of the models of hyperbolic geometry that is very suitable for representing hierarchies due to its exponentially growing volume [25]. The Poincaré ball is defined as an open $n$-ball $\mathbb{D}^n = \{\mathbf{x} \in \mathbb{R}^n : \|\mathbf{x}\| < 1\}$ equipped with a Riemannian metric $g_{\mathbf{x}}^{\mathbb{D}} = \lambda_{\mathbf{x}}^2 g^E$, where $\lambda_{\mathbf{x}} = \frac{2}{1-\|\mathbf{x}\|^2}$, $g^E = \mathbf{I}_n$ is the Euclidean metric tensor, $\lambda_{\mathbf{x}}$ is the *conformal factor*, and $\|\cdot\|^2$ denotes the $L^2$ norm in Euclidean space. The distance between two points $\mathbf{x}, \mathbf{y} \in \mathbb{D}^n$ can be defined by $d_{\mathbb{D}}(\mathbf{x}, \mathbf{y}) = \cosh^{-1}\left(1 + 2\frac{\|\mathbf{x}-\mathbf{y}\|^2}{(1-\|\mathbf{x}\|^2)(1-\|\mathbf{y}\|^2)}\right)$.

**Structured multi-label prediction**   Let $\mathcal{X} \subseteq \mathbb{R}^n$ denote an $n$-dimensional instance space and $\mathcal{L} = \{l_1, l_2, \dots\}, |\mathcal{L}| \geq 2$ denote the finite set of possible labels. Given a set of $N$ training examples $\mathcal{D} = \{(x_i, L_i) \mid 1 \leq i \leq N, x_i \in \mathcal{X}, L_i \subset \mathcal{L}\}$, *multi-label prediction* aims to learn a labeling function $f : \mathcal{X} \to 2^{\mathcal{L}}$ mapping from the instance space to the *powerset* of the label space, $f(x) \subset \mathcal{L}$.

*Structured* multi-label prediction additionally imposes a set of prior-known logical constraints over the labels, namely, the predictions must be logically consistent with these constraints. Analogous to Mirzazadeh et al. [15], we consider two forms of logical constraints between labels: implication and mutual exclusion. Specifically, an *implication* of the form $l_a \Rightarrow l_b$ imposes the constraint that whenever an instance is labeled as $l_a$ then it must also be labeled as $l_b$, i.e., $l_a \Rightarrow l_b$ is a shorthand notation for $\forall x \in \mathcal{X}, l_a \in f(x) \Rightarrow l_b \in f(x)$. *Mutual exclusions* are constraints of the form $\neg l_a \vee \neg l_b$, implying that an instance cannot be simultaneously labeled as $l_a$ and $l_b$, i.e., $\neg l_a \vee \neg l_b$ is a shorthand notation for $\forall x \in \mathcal{X}, l_a \notin f(x) \vee l_b \notin f(x)$. We can concisely represent a set of implication and exclusion constraints with a hierarchy and exclusion (HEX) graph [2].

**Definition 1** (HEX graph [2][2]). *A HEX graph $G = (V, E_h, E_e)$ is a graph consisting of a set of nodes $V = \{v_1, \dots, v_n\}$, directed (hierarchy) edges $E_h \subseteq V \times V$, and undirected (exclusion) edges $E_e \subseteq V \times V$, such that the subgraph $G_h = (V, E_h)$ is a DAG and the subgraph $G_e = (V, E_e)$ has no self loop. Each node $v_i \in V$ represents the label $l_i$. A directed edge $(v_i, v_j) \in E_h$ represents the implication $l_i \Rightarrow l_j$, and an undirected edge $(v_i, v_j) \in E_e$ represents the exclusion $\neg l_i \vee \neg l_j$.*

Note that an arbitrary HEX graph may contain redundant edges. A hierarchy edge $(v_i, v_j)$ is redundant when there is a path in $G_h$ from $v_i$ to $v_j$ which does not contain the edge $(v_i, v_j)$. Similarly, an exclusion edge $(v_i, v_j)$ is redundant when there is another exclusion edge connecting their ancestors (or connecting one node's ancestor to the other node). We can transform a HEX graph into an equivalent HEX graph by adding or removing redundant edges. In this paper, we only consider HEX graphs that have a minimal number of edges, we call such HEX graph a *minimal sparse* HEX graph (see Fig. 1(a) for an example). Given a minimal sparse HEX graph, we define the HEX-property as

**Definition 2** (HEX-property). *A labeling function $f$ has the HEX property with respect to a HEX graph $G$ if for all $x \in \mathcal{X}$, $f(x)$ respects all constraints represented by $G$.*

We also call such function $f$ *logically consistent* w.r.t $G$. Given the HEX graph and the HEX-property, structured multi-label prediction is formally defined as a constrained optimization problem.

**Definition 3** (Structured multi-label prediction). *The structured multi-label prediction task with respect to a training set $\mathcal{D} = \{(x_i, L_i) \mid 1 \leq i \leq N, x_i \in \mathcal{X}, L_i \subset \mathcal{L}\}$, minimal HEX graph $G = (V, E_h, E_e)$, and multi-label prediction function $f$, is the task of learning $f$ such that the function $f$ minimizes $\sum_{(x_i, L_i) \in \mathcal{D}} \mathrm{loss}(f(x_i), L_i)$, with $\mathrm{loss}$ a predefined function, while attempting to maintain the HEX-property with respect to $G$.*

Note that this definition allows for a *soft* interpretation of the constraints, meaning that the goal is to adhere to all of them, but we do allow for loosening some if necessary. For example, a mutual exclusion constraint is allowed to loosen when an instance (e.g., image), though rarely happens, is simultaneously labeled as two mutual exclusive labels (e.g., *dog* and *cat*).

## 3 Hyperbolic embedding inference

We consider learning a real-valued ranking function $h : \mathcal{X} \times \mathcal{L} \mapsto [0, 1]$, where the output is interpreted as the confidence of an instance $x \in \mathcal{X}$ having a label $l \in \mathcal{L}$. Afterward, a binary multi-label classifier $f : \mathcal{X} \to 2^{\mathcal{L}}$ can be simply obtained by thresholding the ranking function with a threshold $t$, i.e., $f(x) = \{l \mid h(x, l) \geq t, \forall l \in \mathcal{L}\}$. The objective of $h$ is to assign higher scores to positive instance-label pairs than that of negative instance-label pairs.

### 3.1 Geometric construction

Given an $n$-dimensional Poincaré ball $\mathbb{D}^n$, we associate each instance $x_i \in \mathcal{X}$ with a point in the Poincaré ball and associate each label $l_i \in \mathcal{L}$ with a Poincaré hyperplane, such that its corresponding positive and negative instances are correctly separated by the hyperplane.

---

[2]Deng et al. [2] use subsumption, which is the inverse relation of implication that we use here.

**Poincaré hyperplanes**  Let $\mathbb{B}^n$ denote the set of $n$-balls in $\mathbb{R}^n$ whose boundaries $\partial\mathbb{B}^n$ intersect the Poincaré ball $\mathbb{D}^n$ perpendicularly. Poincaré hyperplanes are defined by $\partial\mathbb{B}^n \cap \mathbb{D}^n$ (see Fig. 2(a)) plus all linear subspaces going through the origin. For the former cases, a Poincaré hyperplane can be uniquely defined by its center point that has a minimal distance to the origin.

**Definition 4.** *Given a (center) point* $\mathbf{c} \in \mathbb{D}^n$ *where* $\mathbf{c} \neq \mathbf{0}$*, the Poincaré hyperplane is defined as*

$$H_{\mathbf{c}} = \left\{ \mathbf{p} \in \mathbb{D}^n : g^{\mathbb{D}}\left(\log_{\mathbf{c}}\left(\mathbf{p}\right), \vec{\mathbf{c}}\right) = 0 \right\} \tag{1}$$

*where* $\mathbf{c}$ *is the center point and* $\vec{\mathbf{c}} \in T_{\mathbf{c}}\mathbb{D}^n$ [3] *is the normal vector passing through the origin* $\mathbf{0}$*.*

Intuitively, this corresponds to the union of all geodesics passing through $\mathbf{c}$ while orthogonal to the normal vector $\vec{\mathbf{c}} \in T_{\mathbf{c}}\mathbb{D}^n$. In the case where $\mathbf{c}$ is the center of the hyperplane, $\vec{\mathbf{c}}$ must simultaneously pass through $\mathbf{c}$ and the origin. Hence, $\vec{\mathbf{c}}$ can be simply taken as $\mathbf{c}$ without loss of generality. For the special case where $\mathbf{c} = \mathbf{0}$, the Poincaré hyperplanes are all linear subspaces (Euclidean planes) passing through the origin. In this paper, we exclude these special cases by assuming $\mathbf{c} \neq \mathbf{0}$.

**Geometric intuition**  Essentially, the Poincaré hyperplane works as a linear decision boundary that separates the embedding space into two regions,[4] where the smaller region (i.e., convex hull) is interpreted as the space of positive samples while the other one is interpreted as the space of negative samples. Two reasons motivate us to model labels as Poincaré hyperplanes: 1) Modeling labels as hyperplanes has several desired theoretical advantages in margin-based classifiers. Our model shares the same philosophy as existing learning frameworks such as hyperbolic logistic regression [19] and hyperbolic SVM [26]; 2) More importantly, unlike Euclidean space that is flat, hyperbolic Poincaré ball is a curved space in which there are infinitely many non-parallel hyperplanes which do not intersect, implying that linear decision boundaries in hyperbolic space can capture more complicated set-theoretic interactions, such as implication and mutual exclusion.

**Enclosing balls**  Given a Poincaré hyperplane $H_{\mathbf{c}}$, we call the corresponding $n$-ball $\mathbb{B}^n_{\mathbf{c}}$ that encloses $H_{\mathbf{c}}$ its enclosing $n$-ball. Formally, an enclosing $n$-ball $\mathbb{B}^n(\mathbf{o}, r)$ is defined by $\mathbb{B}^n(\mathbf{o}, r) = \{\mathbf{p} : \|\mathbf{p} - \mathbf{o}\| \leq r\}$, where $\mathbf{o} \in \mathbb{R}^n$ and $r$ are the center point and the radius, respectively. Given $H_{\mathbf{c}}$, we have the following closed-form representation of $\mathbb{B}^n_{\mathbf{c}}$.

**Proposition 1.** *Given a Poincaré hyperplane* $H_{\mathbf{c}}$ *where* $\mathbf{c} \neq \mathbf{0}$*, there exists an* $n$*-ball* $\mathbb{B}^n_{\mathbf{c}}(\mathbf{o}_{\mathbf{c}}, r_{\mathbf{c}})$ *such that* $H_{\mathbf{c}} \subset \mathbb{B}^n_{\mathbf{c}}(\mathbf{o}_{\mathbf{c}}, r_{\mathbf{c}})$*, i.e.,* $H_{\mathbf{c}}$ *is a subset of* $\mathbb{B}^n_{\mathbf{c}}(\mathbf{o}_{\mathbf{c}}, r_{\mathbf{c}})$*.* $\mathbb{B}^n_{\mathbf{c}}$ *is uniquely given by*

$$\mathbb{B}^n_{\mathbf{c}} = \mathbb{B}^n\left(\frac{\left(1 + \|\mathbf{c}\|^2\right)}{2\|\mathbf{c}\|}\mathbf{c}, \frac{1 - \|\mathbf{c}\|^2}{2\|\mathbf{c}\|}\right) \tag{2}$$

*Proof sketch.* The key idea is to solve a quadratic equation given by the fact that the radius of $\mathbb{B}^n_{\mathbf{c}}$, the radius of $\mathbb{D}^n$, and the distance from the center of $\mathbb{D}^n$ to the center of $\mathbb{B}^n_{\mathbf{c}}$ must satisfy the Pythagorean theorem [27]. Full proof is in the supplementary material.

### 3.2  Geometric interpretation

Our main idea is to transform the logical relationships between labels into geometric relationships between their corresponding enclosing $n$-balls. In particular, the implication is modeled by the geometric insideness while the mutual exclusion is modeled by the geometric disjointness.

**Implication**  The logical implication between two labels is interpreted as geometric relations between $n$-balls, i.e., $n$-ball insideness illustrated in Fig. 2(b). In particular, an $n$-ball $\mathbb{B}_{\mathbf{w}}(\mathbf{o}_{\mathbf{w}}, r_{\mathbf{w}})$ contains $\mathbb{B}_{\mathbf{u}}(\mathbf{o}_{\mathbf{u}}, r_{\mathbf{u}})$ if and only if $\|\mathbf{o}_{\mathbf{u}} - \mathbf{o}_{\mathbf{w}}\| + r_{\mathbf{u}} < r_{\mathbf{w}}$, and thus we can create an insideness loss defined by

$$\mathcal{L}_{\text{inside}}(\mathbb{B}_{\mathbf{u}}, \mathbb{B}_{\mathbf{w}}) = \max\{0, \|\mathbf{o}_{\mathbf{u}} - \mathbf{o}_{\mathbf{w}}\| + r_{\mathbf{u}} - r_{\mathbf{w}}\}. \tag{3}$$

Clearly, the insideness loss term satisfies the properties of correctness and transitivity

**Lemma 1** (Correctness). $\mathbb{B}_{\mathbf{u}}$ *is inside of* $\mathbb{B}_{\mathbf{w}}$ *if and only if* $\mathcal{L}_{inside}(\mathbb{B}_{\mathbf{u}}, \mathbb{B}_{\mathbf{w}}) = 0$.

**Lemma 2** (Transitivity). *If* $\mathcal{L}_{inside}(\mathbb{B}_{\mathbf{u}}, \mathbb{B}_{\mathbf{w}}) = 0$ *and* $\mathcal{L}_{inside}(\mathbb{B}_{\mathbf{w}}, \mathbb{B}_{\mathbf{v}}) = 0$*, we have* $\mathcal{L}_{inside}(\mathbb{B}_{\mathbf{u}}, \mathbb{B}_{\mathbf{v}}) \leq \mathcal{L}_{inside}(\mathbb{B}_{\mathbf{u}}, \mathbb{B}_{\mathbf{w}}) + \mathcal{L}_{inside}(\mathbb{B}_{\mathbf{w}}, \mathbb{B}_{\mathbf{v}}) \leq \mathcal{L}_{inside}(\mathbb{B}_{\mathbf{w}}, \mathbb{B}_{\mathbf{v}}) = 0$.

---

[3] In this paper, we distinguish normal vectors from regular points by adding an arrow on top of its letters.

[4] Note that by using the metric in the Poincaré ball, each region has infinite (exponentially growing) volume.

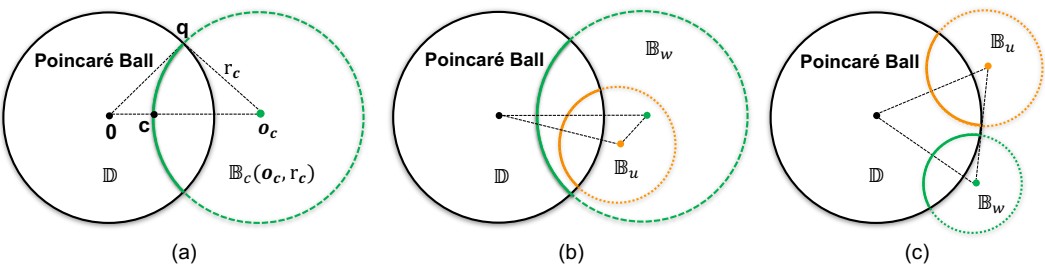

Figure 2: (a) A Poincaré hyperplane is defined as the intersection between the Poincaré ball $\mathbb{D}$ and the boundary of an $n$-ball $\mathbb{B}_\mathbf{c}$. The Poincaré hyperplane is uniquely parameterized by a center point $\mathbf{c}$, and the corresponding $n$-ball (its radius and center) can be uniquely determined by Proposition 1. (b) Label implication is interpreted as $n$-ball insideness. (c) Mutual exclusion is interpreted as $n$-ball disjointedness.

**Mutual exclusion**  Similarly, we interpret mutual exclusion as geometric disconnectedness between $n$-balls illustrated in Fig. 2(c). $\mathbb{B}_\mathbf{u}$ disconnecting from $\mathbb{B}_\mathbf{w}$ can be measured by subtracting the distance between their center points from the sum of their radii. Inversely, the corresponding loss is

$$\mathcal{L}_{\text{disjoint}}(\mathbb{B}_\mathbf{u}, \mathbb{B}_\mathbf{w}) = \max\{0, r_\mathbf{w} + r_\mathbf{u} - \|\mathbf{o}_\mathbf{u} - \mathbf{o}_\mathbf{w}\|\} \tag{4}$$

Again, the disjointedness loss term satisfies the correctness property

**Lemma 3** (Correctness). $\mathbb{B}_\mathbf{u}$ *disconnects from* $\mathbb{B}_\mathbf{w}$ *if and only if* $\mathcal{L}_{\text{disjoint}}(\mathbb{B}_\mathbf{u}, \mathbb{B}_\mathbf{w}) = 0$.

### 3.3 Classification and learning

Given the embeddings of instances and labels, an instance can be classified by measuring the *geometric membership*, i.e., the confidence of a point $\mathbf{p} \in \mathbb{D}^n$ being inside the enclosing ball $\mathbb{B}$.

**Membership and non-membership**  Formally, given an instance embedding $\mathbf{p} \in \mathbb{D}^n$ and a label embedding associated with an enclosing $n$-ball $\mathbb{B}_\mathbf{c}$. The confidence of an instance $\mathbf{p}$ being inside the enclosing $n$-ball $\mathbb{B}_\mathbf{c}$ can be measured by subtracting the distance between the center point of $\mathbb{B}_\mathbf{c}$ and $\mathbf{p}$ from the radius of $\mathbb{B}_\mathbf{c}$. The corresponding loss is defined as the inverse of the measure, given by

$$\mathcal{L}_{\text{membership}}\left(\mathbf{p}, \mathbb{B}_\mathbf{c}\left(\mathbf{o}_\mathbf{c}, r_\mathbf{c}\right)\right) = \max\{0, \|\mathbf{o}_\mathbf{c} - \mathbf{p}\| - r_\mathbf{c}\}. \tag{5}$$

Symmetrically, for negative instance-label relations, the loss of non-membership can be defined as

$$\mathcal{L}_{\text{non-membership}}\left(\mathbf{p}, \mathbb{B}_\mathbf{c}\left(\mathbf{o}_\mathbf{c}, r_\mathbf{c}\right)\right) = \max\{0, r_\mathbf{c} - \|\mathbf{o}_\mathbf{c} - \mathbf{p}\|\}. \tag{6}$$

Clearly, we have the following properties that follow directly from the definitions.

**Lemma 4.** *A point* $\mathbf{p}$ *is a member of* $\mathbb{B}_\mathbf{c}$ *if and only if* $\mathcal{L}_{\text{membership}}\left(\mathbf{p}, \mathbb{B}_\mathbf{c}\right) = 0$.

**Lemma 5.** *A point* $\mathbf{p}$ *is not a member of* $\mathbb{B}_\mathbf{c}$ *if and only if* $\mathcal{L}_{\text{non-membership}}\left(\mathbf{p}, \mathbb{B}_\mathbf{c}\right) = 0$.

**Lemma 1-2, Lemma 3, Lemma 4-5** immediately follow the definitions of geometric insideness, disjointedness, and membership, respectively.

We aim to learn an encoder $E_\theta$ (i.e., a hyperbolic neural network whose designs depend on the datasets), where $\theta$ is the trainable parameter, and a function $\mathcal{C}$ which maps labels to the center points of the corresponding Poincaré hyperplanes in the Poincaré ball.

Now, we define $h(x, l) = \sigma\left(\mathcal{L}_{\text{non-membership}}\left(E_\theta(x), C(l)\right) - \mathcal{L}_{\text{membership}}\left(E_\theta(x), C(l)\right)\right)$, as our ranking function, where $\sigma$ is the sigmoid function. The final classification function is then defined by $f(x) = \{l \mid h(x, l) \geq 0.5\}$. We call our classifier *hyperbolic multi-label embedding inference* (HMI). Given a HEX graph, HMI has the following guarantee.

**Proposition 2** (HEX-property). *The classification function* $f$ *of HMI has the HEX property with respect to* $G$ *if for every constraint in* $G$, *the corresponding loss term is* 0.

**Learning with soft constraints**  Let $\mathcal{D}^+ = \{(x_i, l_n) | (x_i, L_i) \in \mathcal{D}, l_n \in L_i\}$ be the set of positive instance-label pairs and $\mathcal{D}^- = \{(x_i, l_n) | (x_i, L_i) \in \mathcal{D}, l_n \in \mathcal{L}, l_n \notin L_i\}$ be the set of negative instance-label pairs. By combining the loss functions of membership, non-membership, insideness and disjointedness, the learning objective can be formulated as

$$\min_{\theta, \mathcal{C}} \sum_{(x_i, l_n) \in \mathcal{D}^+} \mathcal{L}_{\text{membership}} \left( E_\theta \left( x_i \right), \mathbb{B}_{\mathcal{C}(l_n)} \right) + \sum_{(x_i, l_n) \in \mathcal{D}^-} \mathcal{L}_{\text{non-membership}} \left( E_\theta \left( x_i \right), \mathbb{B}_{\mathcal{C}(l_n)} \right)$$

$$+ \lambda \left( \sum_{(v_i, v_j) \in E_h} \mathcal{L}_{inside} \left( \mathbb{B}_{\mathcal{C}(l_i)}, \mathbb{B}_{\mathcal{C}(l_j)} \right) + \sum_{(v_i, v_j) \in E_e} \mathcal{L}_{disjoint} \left( \mathbb{B}_{\mathcal{C}(l_i)}, \mathbb{B}_{\mathcal{C}(l_j)} \right) \right) \quad (7)$$

The first two terms are losses for positive and negative samples while the last two terms are implication and exclusion constraints, respectively, with $\lambda$ being the penalty weight of the constraints.

The following corollary shows that our model has a strong inductive bias for preserving consistency.

**Corollary 1.** *Given a HEX graph $G$ of labels, if the loss terms $\mathcal{L}_{inside}$ and $\mathcal{L}_{disjoint}$ are $0$, then the learned prediction function is logically consistent.*

**Classification via hyperbolic logistic regression**    A key advantage of our method is that the losses of constraints are compatible with other (margin-based) hyperbolic classifiers such as hyperbolic logistic regression (HLR) [19] and hyperbolic support vector machine (HSVM) [26]. In our experiment we explore HLR, which formulates the *logits* as the distances from an instance to a Poincaré hyperplane of a label. That is, $h(x, l) = d \left( E_\theta \left( x \right), H_C \left( l \right) \right)$. $d(\mathbf{p}, H_\mathbf{c})$ has the following closed form:

$$d(\mathbf{p}, H_\mathbf{c}) = \sinh^{-1} \left( \frac{2 | \langle (-\mathbf{c}) \oplus \mathbf{p}, \mathbf{c} \rangle |}{(1 - \| (-\mathbf{c}) \oplus \mathbf{p} \|^2) \, \| \mathbf{c} \|} \right) \quad (8)$$

where $\oplus$ is the Möbius addition [19]. The classifier is defined by $f(x) = \{ l | \, \sigma \left( h \left( x, l \right) \right) \geq 0.5, \forall l \in \mathcal{L} \}$ where $\sigma$ is the sigmoid function. We dub such classifier combined with HMI as HMI+HLR.

## 4 Evaluation

### 4.1 Experiment setup

**Datasets**    We consider 12 datasets that have been used for evaluating multi-label prediction methods [11, 8, 10]. These consist of 8 functional genomic datasets [28], 3 image annotation datasets [29, 30], and 1 text classification dataset [31]. All input features are pre-processed in the same way as described by Patel et al. [11]. For all datasets, the implication constraints (label taxonomy) are given. Following Mirzazadeh et al. [15] we add exclusion constraints between sibling nodes whenever this does not create a contradiction (i.e., they share no common descendant nodes). We also explore other strategies for deriving exclusions, but no significant difference was observed (see the supplement for an analysis). Similar to MBM [11] and its baselines, we sample 30% of the implications and exclusions constraints for training the model.

**Hyperbolic encoder**    We adopt a simple hyperbolic linear layer as the instance encoder for all datasets. A single-layer hyperbolic fully-forward linear layer is defined by $f_{\theta = \{ \mathbf{W}, \mathbf{b} \}}(\mathbf{x}) = \tanh^\otimes \left( \mathbf{W} \otimes \mathbf{x} \oplus \mathbf{b} \right)$, with $\otimes$ being Möbius matrix-vector multiplication defined by $M \otimes \mathbf{x} = \tanh \left( \frac{\| M\mathbf{x} \|}{\| \mathbf{x} \|} \tanh^{-1}(\| \mathbf{x} \|) \right) \frac{M\mathbf{x}}{\| M\mathbf{x} \|}$, where $\mathbf{W} \in \mathbb{R}^{n \times d}$ is a trainable matrix and $\mathbf{x}$ is a point $\mathbf{x} \in \mathbb{D}^n, M\mathbf{x} \neq 0$. $\oplus$ denotes Möbius addition given by $\mathbf{x} \oplus \mathbf{y} = \frac{\left( 1 + 2 \langle \mathbf{x}, \mathbf{y} \rangle + \| \mathbf{y} \|^2 \right) \mathbf{x} + \left( 1 - \| \mathbf{x} \|^2 \right) \mathbf{y}}{1 + 2 \langle \mathbf{x}, \mathbf{y} \rangle + \| \mathbf{x} \|^2 \| \mathbf{y} \|^2}$. $\tanh^\otimes$ denotes an Möbius version of pointwise non-linearity given by $\tanh^\otimes(\mathbf{x}) = \exp_0 \left( \tanh \left( \log_0(\mathbf{x}) \right) \right)$, with $\exp_0$ and $\log_0$ being the exponential and logarithmic maps, see [19] for more details.

**Baselines**    We compare our approach with both classical vector-based and state-of-the-art region-based embedding methods. In particular, we consider two vector-based models: 1) The multi-label vector model (MVM) [32], which encodes both inputs and labels as Euclidean vectors; 2) the multi-label hyperbolic model (MHM) used by Chen et al. [13], which represents inputs and labels as hyperbolic points; and two box models: 3) the non-probabilistic box model (BoxE) [33] and 4) the probabilistic multi-label box model (MBM) [11] that encodes both instances and labels as axis-parallel hyper-rectangles. Besides, we compare with 5) hyperbolic logistic regression (HLR) [19] since it also encodes labels as Poincaré hyperplanes (but does not use geometric constraints). Furthermore, we

Table 1: Comparison of performance and consistency on 12 datasets, where underline indicates the best results over embedding-based methods, and **boldface** indicates the best results over all methods. We implemented HMI, HLR and HMI+HLR. Other results are taken from Patel et al. [11]. All metrics are averaged across 10 runs with random seeds (standard deviations are relatively small (in range $[2 \times 10^{-4}, 2.3 \times 10^{-3}]$) and are hence omitted).

| Dataset | Metric | Ours | | Embeddings | | | | | Non-embedding |
|---|---|---|---|---|---|---|---|---|---|
| | | HMI | HMI+HLR | MVM | MHM | BoxE | MBM | HLR | C-HMCNN |
| ExprFUN | mAP ↑ | **38.53** | 38.50 | 37.94 | 31.90 | 37.30 | 38.42 | 37.98 | 38.41 |
| | CmAP ↑ | **38.72** | 38.62 | 37.41 | 32.02 | 37.92 | 38.67 | 37.44 | 38.41 |
| | HCV ↓ | 0.92 | 1.07 | 1.97 | 1.92 | 4.79 | 1.87 | 2.17 | **0** |
| CellcycleFUN | mAP ↑ | 34.82 | **34.84** | 31.61 | 28.74 | 31.96 | 34.61 | 34.05 | 34.35 |
| | CmAP ↑ | 34.90 | **35.00** | 31.33 | 28.89 | 32.70 | 34.78 | 34.11 | 34.35 |
| | HCV ↓ | 1.30 | 1.32 | 3.45 | 1.78 | 4.02 | 1.35 | 2.30 | **0** |
| DerisiFUN | mAP ↑ | **36.71** | **36.71** | 24.16 | 24.40 | 26.66 | 28.71 | 26.65 | 28.19 |
| | CmAP ↑ | **36.94** | 36.89 | 24.35 | 24.52 | 26.96 | 28.88 | 26.83 | 28.19 |
| | HCV ↓ | 0.73 | 0.87 | 4.01 | 0.85 | 2.27 | 1.43 | 2.30 | **0** |
| SpoFUN | mAP ↑ | **36.47** | 36.44 | 24.21 | 26.57 | 27.97 | 29.62 | 28.29 | 29.18 |
| | CmAP ↑ | 36.43 | **36.54** | 24.55 | 26.79 | 28.38 | 29.78 | 28.31 | 29.18 |
| | HCV ↓ | 0.92 | 1.05 | 4.73 | 1.69 | 2.75 | 1.53 | 1.98 | **0** |
| ExprGO | mAP ↑ | 48.63 | 48.50 | 44.97 | 40.52 | 46.75 | 48.45 | 48.65 | 48.61 |
| | CmAP ↑ | 48.68 | 48.61 | 41.84 | 40.70 | 47.28 | 48.56 | 48.65 | 48.61 |
| | HCV ↓ | 1.37 | 1.45 | 7.05 | 5.19 | 5.74 | 1.91 | 1.35 | **0** |
| CellcycleGO | mAP ↑ | 45.58 | 45.51 | 44.19 | 39.74 | 43.08 | 44.93 | 40.28 | **45.61** |
| | CmAP ↑ | 45.58 | 45.53 | 41.02 | 39.76 | 43.79 | 45.01 | 40.30 | **45.61** |
| | HCV ↓ | 1.19 | 1.12 | 3.03 | 2.49 | 5.06 | 2.16 | 3.26 | **0** |
| DerisiGO | mAP ↑ | **42.31** | 42.12 | 41.13 | 40.10 | 40.44 | 42.02 | 40.33 | 42.24 |
| | CmAP ↑ | **42.38** | 42.28 | 38.21 | 40.20 | 40.73 | 42.12 | 40.35 | 42.24 |
| | HCV ↓ | 0.86 | 0.99 | 3.46 | 2.02 | 3.16 | 1.13 | 2.31 | **0** |
| SpoGO | mAP ↑ | 42.70 | 42.74 | 42.20 | 39.70 | 40.88 | 41.74 | 39.22 | **42.77** |
| | CmAP ↑ | 42.76 | 42.77 | 39.04 | 39.77 | 41.27 | 41.54 | 39.26 | **42.77** |
| | HCV ↓ | 0.95 | 1.20 | 2.77 | 1.90 | 3.89 | 1.80 | 2.33 | **0** |
| Enron | mAP ↑ | 80.43 | 80.43 | 73.68 | 75.62 | **80.44** | 80.06 | 78.87 | 80.04 |
| | CmAP ↑ | 80.50 | 80.47 | 66.87 | 75.68 | 80.46 | 80.05 | 78.94 | 80.04 |
| | HCV ↓ | **0** | **0** | 2.53 | 0.36 | 0.20 | 0.03 | 0.04 | **0** |
| Diatoms | mAP ↑ | **79.19** | 79.10 | 72.65 | 56.86 | 43.71 | 79.14 | 77.90 | 76.23 |
| | CmAP ↑ | **79.40** | 79.36 | 72.18 | 56.07 | 45.16 | 79.23 | 78.07 | 76.23 |
| | HCV ↓ | 0.17 | 0.18 | 19.20 | 5.55 | 6.39 | 0.34 | 6.36 | **0** |
| Imclef07a | mAP ↑ | **90.67** | 89.60 | 78.22 | 65.30 | 83.71 | 69.26 | 88.33 | 90.26 |
| | CmAP ↑ | **90.89** | 89.71 | 77.46 | 66.01 | 84.73 | 69.48 | 88.45 | 90.26 |
| | HCV ↓ | 0.20 | 0.19 | 22.86 | 4.75 | 12.73 | 2.40 | 1.77 | **0** |
| Imclef07d | mAP ↑ | 89.19 | 89.20 | 88.59 | 75.69 | 87.95 | 89.56 | 88.91 | 89.22 |
| | CmAP ↑ | 90.00 | 90.02 | 86.87 | 76.95 | 88.93 | 90.07 | 87.38 | 89.22 |
| | HCV ↓ | 0.37 | 0.36 | 11.02 | 7.56 | 11.93 | 5.66 | 6.88 | **0** |
| **Avg. Rank ↓** | mAP | 1.75 | 2.42 | 6.33 | 7.58 | 5.75 | 3.5 | 5.25 | 3.08 |
| | CmAP | 1.58 | 2.08 | 7.16 | 7.41 | 5.25 | 3.58 | 5.41 | 3.33 |
| | HCV | 2.25 | 2.75 | 7.42 | 5.25 | 7.25 | 4.25 | 5.58 | **1.00** |

compare with 6) C-HMCNN, a state-of-the-art non-embedding based method that injects hierarchy constraints directly into the loss function without embedding labels. A notable difference is that C-HMCNN needs the full hierarchy constraints as its input. Finally, we also implement HMI+HLR, a combination of our proposed constraints with HLR for an ablation study.

**Implementation details** We implement HMI, HLR and HMC-HLR using PyTorch [34] and train the models on NVIDIA A100 with 40GB memory. We train HMI, HLR and HMI+HLR using Riemannian Adam [35] optimizer implemented by the Geoopt library [36] with a batch size of 4. We also explore some larger batch sizes but it does not yield better results, which is also observed in Wehrmann et al.[14]. We set the dropout rate to 0.6 suggested by [14] to avoid the case that the model overfits the small training sets. We employ an early-stopping strategy with patience 20 to save training time. The results of other baselines are as reported by Patel et al.[11] that we closely follow. The learning rate is searched from $\{1e-4, 5e-4, 1e-3, 5e-3, 1e-2\}$. The penalty weight of the violation is searched from $\{1e-5, 5e-4, 1e-4, 5e-3, 1e-2\}$ and we also show its impact in an ablation. The best dimension per dataset is searched from $\{32, 64, 128, 256\}$, which is one order of magnitude lower than that used by Patel et al. [11] ($\{250, 500, 1000, 1750\}$). All methods have been run 10 times with random seeds and the average results are reported. We omit the standard deviations since they are in a very small range ($[2 \times 10^{-4}, 2.3 \times 10^{-3}]$). Our code is open available at [5].

---
[5]https://github.com/xiongbo010/HMI

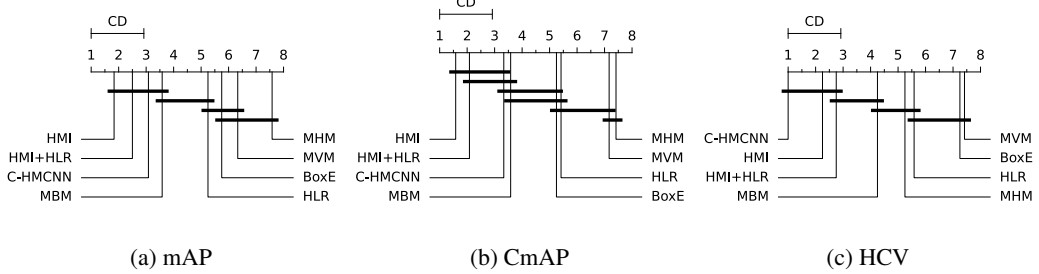

Figure 3: Critical diagrams of the post-hoc Nemenyi test across all 12 datasets.

**Evaluation protocols**  In line with Patel et al. [11], we consider *Mean Average Precision (mAP)*,[6] which summarizes the information of precisions and recalls with varied thresholds. We also report two metrics that additionally take the constraints into account: 1) Constrained mAP (CmAP) is a variant of mAP that replaces the score of each label with the maximum scores of its descendant labels in the hierarchy [11]. 2) *Hierarchy Constraint Violation (HCV)* [11] measures the extent to which the label scores violate the implication constraints regardless of true labels for the instances. HCV is computed as $\text{HCV} = \frac{1}{|\mathcal{D}||E_h|} \sum_{k=1}^{|\mathcal{D}|} \sum_{(l_i, l_j) \in E_h} \mathbb{1} \left( h_i^k - h_j^k > 0 \right)$, where $h_i$ means the prediction score of label $l_i$. Clearly, a lower value of HCV implies higher consistency in the predictions.

## 4.2  Main results

As Table 1 shows, our method HMI either achieves the best (7-8/12 datasets) or competitive (4-5/12 datasets) performance (mAP and CmAP) over all compared methods. HMI outperforms all methods w.r.t the average ranking of mAP/CmAP, showcasing the advantages of HMI. We observed that the CmAP is close to mAP, indicating that the model is adhering to the label constraints [11]. In terms of predictive consistency (HCV), HMI consistently achieves the best or the second-best results. Note that C-HMCNN always gets zero HCV because it exploits the complete hierarchy. HMI achieves competitive HCV, despite only using 30% of the hierarchy.

**Statistical significance**  Following Patel et al. [11] and Giunchiglia and Lukasiewicz [8], we test the statistical significance of the performance across all datasets. First, we perform the Friedman test [37] and show that there exists a significant difference w.r.t. all metrics with p-values $\ll 0.05$.

Table 2: Results of Wilcoxon test over HMI against baselines.

| Method | mAP | CmAP | CV |
|---|---|---|---|
| HMI vs C-HMCNN | $5.8 \times 10^{-4}$ | $4.4 \times 10^{-4}$ | $5.0 \times 10^{-3}$ |
| HMI vs MBM | $3.3 \times 10^{-4}$ | $2.4 \times 10^{-4}$ | $4.9 \times 10^{-4}$ |
| HMI vs HMI+HLR | $2.3 \times 10^{-2}$ | $3.8 \times 10^{-2}$ | $9.7 \times 10^{-1}$ |

Next, we conduct the post-hoc Nemenyi test to verify the statistical differences of the average ranking. The critical diagram w.r.t the average ranking of mAP/CmAP is shown in Fig. 3, in which the methods that have no significant differences (significance level 0.05) are connected by a horizontal line. As shown in the diagrams, it is clear to conclude that there is a statistically significant difference w.r.t mAPs/CmAPs of HMI and HMI+HLR against MVM, BoxE, MHM, and HLR but not the two strong baselines (MBM and C-HMCNN). We further perform the Wilcoxon test that considers not only the differences in rankings but also the numerical differences in the performance. The Wilcoxon test results show that there is a statistically significant difference between the mAPs/CmAPs of HMI and the two strong baselines with p-value $\ll 0.05$. In terms of HCV, our statistical significance test in Figure 3 and Table 2 shows that HMI and HMI+HLR significantly outperform MVM, BoxE, MHM, HLR, and MBM but not C-HMCNN since it has zero HCV. However, we observed that the predictive performance (mAP, CmAP) is not fully proportional to the HCV, e.g., HMI outperforms C-HMCNN w.r.t. mAP/CmAP on many of the datasets even though C-HMCNN has zero CV.

---

[6]https://scikit-learn.org/stable/modules/generated/sklearn.metrics.average_precision_score.html

**Classification via hyperbolic logistic regression**   To validate whether our proposed geometric constraints are able to improve hyperbolic logistic regression (HLR) [19], we implement HMI+HLR, a combination of our proposed constraints with HLR as described in Section 3.3. Table 1 show that HMI+HLR outperforms HLR with statistical confidence, showcasing that HMI is able to improve the predictive performance and consistency of HLR. However, there is no significant difference (with $p$-value larger than 0.05 in Table 2) between the two variants of our method (HMI and HMI+HLR).

## 4.3  Ablation studies & parameter sensitivity.

For further ablation, we introduce one additional metric. Exclusion Constraint Violation (ECV) measures, analogous to HCV, the fraction of the exclusion constraints violated by the predictions i.e., $\text{ECV} = \frac{1}{|\mathcal{D}||E_e|} \sum_{k=1}^{|\mathcal{D}|} \sum_{(l_i,l_j)\in E_e} \mathbb{1}\left(f_i^k \wedge f_j^k\right)$. We introduce this because HCV can be made zero trivially by associating all labels with the same score. Hence, in the ablation study, we will show how the exclusion constraints (the results of ECV) complement HCV and influence the overall performance.

**Impact of penalty weight**   Table 3 shows the results of HMI on "CellcycleFUN" and "Cellcy­cleGO" dataset. We observed that with different penalty weights, the obtained results are slightly differ­ent. Even without penalty ($\lambda = 0$), the model already achieves ac­ceptable results, in particular, it outperforms MVM, MHM, and BoxE, indicating that our hyper­bolic model, to some extent, is ca­

Table 3: Impact of violation penalty weight $\lambda$ on CellcycleFUN and CellcycleGO dataset.

| Dataset | Metric | $\lambda = 0.0$ | $\lambda = 0.001$ | $\lambda = 0.005$ | $\lambda = 0.01$ | $\lambda = 0.1$ |
|---|---|---|---|---|---|---|
| CellcycleFUN | mAP | 33.87 | 34.78 | **34.82** | 34.76 | 32.28 |
|  | CmAP | 34.03 | 34.83 | **34.90** | 34.85 | 33.75 |
|  | HCV | 2.33 | 1.87 | 1.30 | 1.04 | **0.75** |
|  | ECV | 4.33 | 3.77 | 2.40 | 1.67 | **1.35** |
| CellcycleGO | mAP | 40.26 | 41.47 | **45.58** | 45.56 | 41.28 |
|  | CmAP | 39.87 | 42.05 | 45.58 | **45.60** | 40.75 |
|  | HCV | 2.28 | 1.57 | 1.19 | 0.99 | **0.86** |
|  | ECV | 3.98 | 3.27 | 2.17 | 1.71 | **1.34** |

pable of capturing label hierarchies without any explicit constraints. However, as Table 3 shows, a proper $\lambda = 0.001$, $\lambda = 0.005$ and $\lambda = 0.01$ indeed improves the performance and consistency. Finally, we observed that increasing $\lambda$ to 0.1, though further improves consistency (HCV and ECV), does not further improve mAP and CmAP. We conjecture that this is because a large $\lambda$ would encourage the model to "overfit" the given constraints while "underfitting" the classification loss.

**Impact of implication & exclu­sion**   To study the roles of impli­cation and exclusion. We imple­mented three variants of HMI by removing either implication or ex­clusion, or removing both of them. Table 4 depicts the results of these variants. It is clear that both impli­cation and exclusion constraints improve the base model that has no constraints.   When implica­tion and exclusion are jointly con­

Table 4: Impact of implication and exclusion constraints on Cellcycle­FUN and CellcycleGO dataset.

| Dataset | Metric | HMI | w/o implication | w/o exclusion | non constraints |
|---|---|---|---|---|---|
| CellcycleFUN | mAP | **34.82** | 34.70 | 34.74 | 33.87 |
|  | CmAP | **34.90** | 34.75 | 34.82 | 34.03 |
|  | HCV | 1.30 | 2.34 | 1.45 | 2.33 |
|  | ECV | 2.40 | 2.67 | 3.63 | 4.33 |
| CellcycleGO | mAP | **45.58** | 42.56 | 44.50 | 40.26 |
|  | CmAP | **45.58** | 42.56 | 45.31 | 39.87 |
|  | HCV | 1.19 | 2.16 | 1.73 | 2.28 |
|  | ECV | 2.17 | 3.68 | 3.07 | 3.98 |

strained, the performance is significantly improved again. We also observed that implication and exclusion constraints, to some extent, do complement each other, e.g., by only using implication (resp. exclusion), the model archives lower ECV (resp. HCV). Finally, we observed that even without exclusion, our model still slightly outperforms MBM, showcasing the advantages of hyperbolic space for modeling hierarchies.

**Impact of sampling ratio**   To study whether our method is able to preserve logical constraints from incomplete label constraints we compare the performance of HMI with different ratios for sampling the training constraints. As Figure 4(a) depicts, with zero sampling ratio, our method already achieves acceptable results. We conjecture that this is because some constraints can be learned from the data. However, Figure 4(a) clearly shows that including constraints indeed helps to improve the performance. Making the sampling ratio larger than 30-40% does not lead to a significant

performance gain. We conjecture that this is because certain ratio of training constraints is sufficient for inferring the full set of constraints.

**Impact of embedding dimensionality** We study how the choice of dimensionality affects performance. As Figure 4(b) depicts, HMI achieves acceptable results even in a very low dimension ($n \leq 100$). When increasing the dimension an order of magnitude ($n = 1000$), the performance grows only slightly. Note that all reported baselines achieved acceptable results with dimensions in $[500, 1000, 1750]$ (see

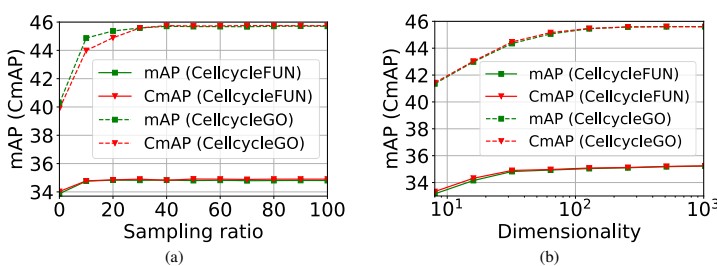

Figure 4: (a) The variation of performance w.r.t the sampling ratio. (b) The variation of performance w.r.t the embedding dimensions.

hyperparameter settings in the Appendix of Patel et al. [11]). We conjecture that the reason we can achieve good performance with fewer dimensions is that the hyperbolic hyperplane is more suitable for representing hierarchical decision boundaries.

**Comparison with MBM with only implication or without any constraint** To faithfully study the advantages of hyperbolic hyperplane on modeling label relations than that of the box model (MBM), we also implement two versions of HMI by considering only (30%) implication constraints and without any constraint (sampling ratio= 0), respectively. Our Wilcoxon test in Table 5 shows that HMI with only implication and HMI without any constraint still outperform their corresponding counterparts of MBM on CmAP and HCV (with p-value $< 0.05$) while achieving comparable results on mAP (i.e., with better average ranks but without statistical significance, we believe this is because mAP is less sensitive to the constraints than CmAP).

Table 5: Results of Wilcoxon test on HMI against MBM in the settings where only implications are available and without any constraint. $-$ means no statistical difference between the compared methods.

| Method | mAP | CmAP | CV |
|---|---|---|---|
| HMI (impl.) vs MBM (impl.) | $-$ | $2.4 \times 10^{-4}$ | $1.2 \times 10^{-3}$ |
| HMI (no conts.) vs MBM (no conts.) | $-$ | $1.3 \times 10^{-2}$ | $6.1 \times 10^{-3}$ |

## 5 Conclusion

In this paper, we focus on a structured multi-label prediction task whose output is supposed to respect the implication and exclusion constraints. We show that such a problem can be formulated in a hyperbolic Poincaré ball space whose linear decision boundaries (Poincaré hyperplanes) can be interpreted as convex regions. The implication and exclusion constraints are geometrically interpreted as insideness and disjointedness, respectively. Experiments on 12 datasets show significant improvements in mean average precision and lower constraint violations, even with an order of magnitude fewer dimensions than baselines.

## Acknowledgments

The authors thank the International Max Planck Research School for Intelligent Systems (IMPRS-IS) for supporting Bo Xiong. Bo Xiong is funded by the European Union's Horizon 2020 research and innovation programme under the Marie Skłodowska-Curie grant agreement No: 860801. Mojtaba Nayyeri is funded by the German Federal Ministry for Economic Affairs and Climate Action under Grant Agreement Number 01MK20008F (Service-Meister). We thank Prof. Martin Keller-Ressel and all anonymous reviewers for providing valuable suggestions that improved the paper.

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
