# A  Proof of theorems

**Proposition 1.** *Given a Poincaré hyperplane $H_{\mathbf{c}}$ where $\mathbf{c} \neq \mathbf{0}$, there exists an $n$-ball $\mathbb{B}_{\mathbf{c}}\left(\mathbf{o}_{\mathbf{c}}, r_{\mathbf{c}}\right)$ such that $H_{\mathbf{c}} \subset \mathbb{B}_{\mathbf{c}}\left(\mathbf{o}_{\mathbf{c}}, r_{\mathbf{c}}\right)$, i.e., $H_{\mathbf{c}}$ is a subset of $\mathbb{B}_{\mathbf{c}}\left(\mathbf{o}_{\mathbf{c}}, r_{\mathbf{c}}\right)$. $\mathbb{B}_{\mathbf{c}}$ is uniquely given by*

$$\mathbb{B}_{\mathbf{c}}^{n} = \mathbb{B}^{n}\left(\frac{\left(1 + \|\mathbf{c}\|^2\right)}{2\|\mathbf{c}\|}\mathbf{c}, \frac{1 - \|\mathbf{c}\|^2}{2\|\mathbf{c}\|}\right) \tag{9}$$

*Proof.* Since $c$ is the center point of the Poincaré hyperplane, the vector $\overrightarrow{c}$ must be a normal vector of the tangent space $T_c\mathbb{B}^n$ of $\mathbb{B}^n$ at $c$. Let $q$ be one of the point that the Poincaré hyperplane and the Poincaré ball intersect at. Then, the radius of $\mathbb{B}_{\mathbf{c}}\left(\mathbf{o}_{\mathbf{c}}, r_{\mathbf{c}}\right)$, the radius of $\mathbb{D}^n$, and the distance from the centers of $\mathbb{D}^n$ to the center of $\mathbb{B}_{\mathbf{c}}\left(\mathbf{o}_{\mathbf{c}}, r_{\mathbf{c}}\right)$ must satisfy the Pythagorean theorem [27], i.e., the three Euclidean distances $d(\mathbf{0}, q)$, $d(q, \mathbf{o}_{\mathbf{c}})$ and $d(\mathbf{o}_{\mathbf{c}}, \mathbf{0})$ must satisfy

$$d(\mathbf{0}, \mathbf{q})^2 + d(\mathbf{q}, \mathbf{o}_{\mathbf{c}})^2 = d(\mathbf{o}_{\mathbf{c}}, \mathbf{0})^2 = \left(d\left(\mathbf{0}, \mathbf{c}\right) + d\left(\mathbf{c}, \mathbf{o}_{\mathbf{c}}\right)\right)^2. \tag{10}$$

Since we have $d\left(\mathbf{c}, \mathbf{o}_{\mathbf{c}}\right) = d(\mathbf{q}, \mathbf{o}_{\mathbf{c}}) = r_{\mathbf{c}}$, by solving this quadratic equation, we have $r_{\mathbf{c}} = \frac{1 - \|\mathbf{c}\|^2}{2\|\mathbf{c}\|}$. Since $\mathbf{o}_{\mathbf{c}} = \mathbf{c}(1 + \frac{r_{\mathbf{c}}}{d(\mathbf{0}, \mathbf{c})})$, we have $\mathbf{o}_{\mathbf{c}} = \mathbf{c}\frac{\left(1 + \|\mathbf{c}\|^2\right)}{2\|\mathbf{c}\|}$. Thus, $\mathbb{B}_{\mathbf{c}} = \mathbb{B}\left(\mathbf{o}_{\mathbf{c}} = \mathbf{c}\frac{\left(1 + \|\mathbf{c}\|^2\right)}{2\|\mathbf{c}\|}, r_{\mathbf{c}} = \frac{1 - \|\mathbf{c}\|^2}{2\|\mathbf{c}\|}\right)$. $\qquad\square$

**Proposition 2** (HEX-property). *The classification function $f$ has the HEX property with respect to $G$ if and only if for any constraint in $G$, the corresponding loss term is $0$.*

*Proof.* Note that the loss term of the constraint being $0$ implies that the corresponding constraint is respected. Our loss terms clearly connect the HEX property. That is, for any point $\mathbf{p} \in D^n$ and a pair of enclosing $n$-balls $(\mathbb{B}_w, \mathbb{B}_u)$, $\mathcal{L}_{\text{membership}}\left(p, \mathbb{B}_w\right) \geq \mathcal{L}_{\text{membership}}\left(p, \mathbb{B}_u\right)$ for all $(\mathbb{B}_w, \mathbb{B}_u)$ where $\mathcal{L}_{\text{inside}}(\mathbb{B}_w, \mathbb{B}_u) = 0$ and $\neg\mathcal{L}_{\text{membership}}\left(p, \mathbb{B}_w\right) \vee \neg\mathcal{L}_{\text{membership}}\left(p, \mathbb{B}_u\right)$ for all $(\mathbb{B}_w, \mathbb{B}_u)$ where $\mathcal{L}_{\text{disjoint}}(\mathbb{B}_u, \mathbb{B}_w) = 0$. According to the definition of HEX-property, $f$ has the HEX property with respect to $G$ if and only if the corresponding loss term of the corresponding constraint is $0$. $\qquad\square$

**Corollary 1.** *Given a HEX graph $G$ of labels and if the loss of the embeddings is $0$, then the learned prediction function is logically consistent with respect to $G$.*

*Proof.* Note that the loss terms $\mathcal{L}_{\text{inside}}, \mathcal{L}_{\text{disjoint}}, \mathcal{L}_{\text{membership}}, \mathcal{L}_{\text{non-membership}}$ in Eq.7 are all non-negative. Hence, the loss being $0$ implies that all losses are zeros (all constraints are satisfied). According to the definition of consistency, the prediction function is consistent. $\qquad\square$

# B  Supplementary experiments and details

**Datasets and pre-processing**    The functional genomic datasets (Expr, Spo, Derisi, Cellcycle) are available at [7]. The image datasets (Imclef07a, Imclef07d, Diatoms) and text dataset (Enron) are all available at [8]. All licenses of the datasets can be found in the corresponding links and references. The number of labels, types of features, the number of instances vary significantly. The diversity of these datasets makes them suitable for evaluating the multi-label classification task. The input features are pre-processed in the same way as described in [11, 8, 10]. In particular, all categorical features were transformed using one-hot encoding. The missing values were replaced by the mean value (for numeric features) or zero-valued vector (for categorical features). All continuous features were standardized before feeding into the encoder. The labels of the root nodes are removed from training and evaluation.

---

[7]https://dtai.cs.kuleuven.be/clus/hmcdatasets/
[8]http://kt.ijs.si/DragiKocev/PhD/resources/doku.php?id=hmc_classification

Table 6: Statistical information of the datasets used in experiments. Number of features (F), number of classes (L), and number of instances for each dataset split.

| Dataset | Domain | Feature | Label | #Label | #Train | #Val | #Test |
|---------|--------|---------|-------|--------|--------|------|-------|
| ExprFUN | Genomics | Continuous | Forest | 500 | 1636 | 849 | 1288 |
| CellcycleFUN | Genomics | Continuous | Forest | 500 | 1628 | 848 | 1281 |
| DerisiFUN | Genomics | Continuous | Forest | 500 | 1608 | 842 | 1275 |
| SpoFUN | Genomics | Continuous | Forest | 500 | 1600 | 837 | 1266 |
| ExprGO | Genomics | Continuous | DAG | 4132 | 1636 | 849 | 1288 |
| CellcycleGO | Genomics | Continuous | DAG | 4126 | 1625 | 848 | 1278 |
| DerisiGO | Genomics | Continuous | DAG | 4120 | 1605 | 842 | 1272 |
| SpoGO | Genomics | Continuous | DAG | 4120 | 1597 | 837 | 1263 |
| Diatoms | Image | Continuous | Tree | 399 | 1500 | 565 | 1054 |
| Imclef07a | Image | Continuous | Tree | 97 | 7000 | 3000 | 1006 |
| Imclef07d | Image | Continuous | Tree | 47 | 7000 | 3000 | 1006 |
| Enron | Text | Binary | Tree | 57 | 650 | 338 | 600 |

Table 7: The number of exclusion edges derived from the label taxonomy ($A$) and the label co-occurrence ($B$).

| Dataset | A | B |
|---------|---|---|
| ExprFun | 110958 | 110941 |
| CellcycleFUN | 110959 | 110942 |
| DerisiFUN | 111009 | 110992 |
| SpoFUN | 111008 | 110991 |
| ExprGO | 8305590 | 8310506 |
| CellcycleGO | 8305590 | 8310506 |
| SpoGO | 8257458 | 8262341 |
| Diatoms | 78793 | 78799 |
| Enron | 965 | 965 |
| ImCLEF07A | 4417 | 4425 |
| ImCLEF07D | 979 | 985 |

**Deriving mutual exclusion** In real-world applications, exclusion relations could be annotated by human experts by exploiting domain knowledge. In this paper, we explore various strategies to generate possible exclusion relations: 1) *Deriving exclusion from the label taxonomy*. Following the "exclusive whenever possible" assumption [1], we add mutual exclusion edges between two nodes whenever they do not share any descendant nodes (i.e., it does not create a contradiction). 2) *Deriving exclusion from the label co-occurrence*. We add mutual exclusion edges between two labels whenever there is no instance in the training set simultaneously belonging to them. Clearly, strategy 1 generates all possible exclusion edges entailed by the label taxonomy, while strategy 2 generates exclusion edges that are reflected by the dataset itself. Strategy 1 might create false positive exclusions (i.e., exclusions that violate the label co-occurrence in the datasets), while strategy 2 might suffer from the noisy labeled data (e.g., an instance might be incorrectly or incompletely labeled). However, Table 7 shows that there is no statistical difference between the generated exclusions from these two methods. Hence, we may conclude that the "exclusive whenever possible" assumption almost holds. One common problem of these two methods is that there are many redundant edges generated. To efficiently exploit the constraints, we only generate exclusions between sibling nodes whenever it does not create contradiction [15].