# OpenReview forum: "Hyperbolic Embedding Inference for Structured Multi-Label Prediction"
_NeurIPS.cc/2022/Conference — NeurIPS 2022 Accept_

### Official Review · Reviewer_UtNv · 2022-06-29

**Rating:** 6
**Confidence:** 3
**Soundness:** 3 good
**Presentation:** 3 good
**Contribution:** 3 good

**Summary:**

This paper introduces a method for multi-label classification when the class labels have known dependency structures, namely implication and mutual exclusion. Hyperbolic geometry is employed to jointly learn parameters for an encoder that embeds points in a Poincare ball and Poincare hyperplanes for class labels. If the embedded point lies on one side ("inside") of the Poincare hyperplane, it is predicted to have that class label. Due to the curvature of the space, the Poincare hyperplanes can be contained completely inside one another, overlap somewhat, or non-overlapping. The method presented in this work presents a joint training objective to not only correctly classify the examples, but also encourage known constraints to be satisfied by the learned Poincare hyperplanes for each class: implication <-> complete containment and mutual exclusion <-> non-overlapping. The method is compared on standard benchmark datasets against reasonable baselines.

**Questions:**

- Are there reasonable probabilistic semantics for Poincare hyperplanes representing probabilities of events? One appealing quality of box embeddings is their natural probabilistic interpretation and closure under intersection, etc. Does your approach have an analogous interpretation?
- Do the volumes of the "membership regions" correlate with the frequency of the respective class label?
- Is there a way to represent (un)certainty of the given constraints? Say from a not always 100% reliable taxonomy.
- If you train your model with only a fraction of the available constraints, does the resulting set of Poincare hyperplanes for each class recover any of the constraints that weren't used for training? Furthermore, are there any constraints that the resulting Poincare hyperplanes imply that are incorrect?
- Are the constraints provided for training always satisfied by the learned Poincare hyperplanes?
- It seems like the method presented performs much worse when no constraints are provided as compared to MBM. Is there a reason why HMI performs so much worse than MBM when both models are not provided with any constraints? (Context: Table 4 from the submission and Table 1 from Patel et al.)
- Why is it that the method presented can get away with using substantially fewer dimensions? Is this due to the geometry of the Poincare ball? Also, I'm not sure that the dimensionality claim is valid or sound (See Appendix G of Patel et al. --> This shows similar results to Figure 4b in the submission).


**Limitations:**

The authors have adequately addressed the limitations of their work.
The potential negative societal impact of their work is not addressed, but their work has no more potential negative societal impact than any other paper submitted to NeurIPS.

**Strengths And Weaknesses:**

Strengths:
- Paper is overall well-written
- Motivation is clear
- Figures 1 & 2 make the method very clear and easy to understand!
- Experimental evaluation is adequate and convincing with several reasonable ablations

Weaknesses:
- It seems like the method is narrowly applicable. It only seems to beat the best baseline methods when constraints are available. How often are these constraints actually available in practice?
- As presented, only two types of logical constraints can be (softly) enforced by the objective. These constraints are arguably the most common types of constraints for these problems though.
- The claim that their method uses less dimensionality than baselines is somewhat misleading (See Appendix G of Patel et al.)
- The authors don't compare their method to the best performing baseline MBM without explicit constraint modeling.

---

> ### Author Response · Authors · 2022-08-02
> **Response to Reviewer UtNv**
>
> **It seems like the method is narrowly applicable. It only seems to beat the best baseline methods when constraints are available. How often are these constraints actually available in practice?**
> Introducing background knowledge to machine learning problems has many real-world applications. In general, such background knowledge can be obtained either from large-scare knowledge bases such as Wikipedia and domain ontologies or by additional knowledge engineering techniques (e.g., techniques of taxonomies and exclusion derivation). For example, there are many scenarios where these constraints are already available such as functional genomics (e.g., protein function prediction) where the constraints can be obtained from Gene Ontology.
>
> Note that with partial constraints or even without any constraint, HMI is able to recover the taxonomy information. See our new comparison of HMI against MBM where only implication or no any constraint is available in our updated Appendix. Our new results show statistically significant improvement on CmAP and CV measure while achieving comparable results on mAP.
>
> **As presented, only two types of logical constraints can be (softly) enforced by the objective.** More complex constraints such as existing rules with roles $\exists C.r \sqsubset D$ can be encoded but it requires additional encoding such as embeddings of roles (or relations), which has been studied in ontology and knowledge graph embeddings [1], but is less often encountered in real-world settings. We believe this is an interesting trade-off problem and leave it for future research.
>
> [1] Maxat Kulmanov, et al. EL Embeddings: Geometric Construction of Models for the Description Logic EL++. IJCAI 2019.
>
> **The authors don't compare their method to the best performing baseline MBM without explicit constraint modeling.** Thank you for this comment. We have added such comparison (See our response to Reviewer zjFa (Q1) ).
>
> **Probabilistic semantics for Poincare hyperplanes?** Our method does have probability interpretation as in box embedding. For example, the probability of the subclass relation is represented by one ball being inside another ball to a larger or lesser extent.
>
> **Closure under intersection.** We agree that intersectional closure is a nice property in general. However, it also limits the expressiveness for embedding hierarchy and exclusion (HEX) graph. A counter example (Fig. 5 in the Appendix B) is given in the Appendix B (proof is given by [20]). The major issue of box embeddings is that the lower-way intersection may enforce higher-way intersection (e.g., pair-wise two-way intersections of three boxes enforce three-way intersection of the three boxes) because of intersectional closure,, leading to the fact that the box model is not able to represent some simple HEX graphs in any dimension. This is a phenomenon we have observed in our used dataset, e.g., in functional genomic datasets, where many proteins functions (which are labels) have pair-wise (lower-way) intersections but a group of them are not intersected.
>
> **Do the volumes of the "membership regions" correlate with the frequency of the respective class label?**  Yes, see our response to Reviewer mG3n (Q1) with an empirical validation.
>
> **Is there a way to represent (un)certainty of the given constraints? Say from a not always 100\% reliable taxonomy.**  Yes, as we explained, our embedding has probability interpretation so it is possible to deal with (un)certainty.
>
> **If you train your model with only a fraction of the available constraints, does the resulting set of Poincare hyperplanes recover any of the constraints that weren't used for training?**
> Yes, this can be observed from our evaluation (Table 1 and Fig 4a). HMI uses partial (30\%) constraints as training, but achieves very low constraint violation (even $0$ violation in Eron), relecting that the constraints are recovered.
>
> **Are the constraints provided for training always satisfied by the learned Poincare hyperplanes?**
> "Constraints provided for training are always satisfied" means zero training loss of insideness and disjointedness. In some datasets like Enron, we did get almost zero loss with $0$ constraint violation (see result of Enron in Table 2). But in general, the loss is not necessarily to be $0$ (e.g., there might be some inconsistencies of the labeling of training data).
>
> **Why substantially fewer dimensions?**  Yes, we believe that this is because hyperbolic space has exponentially growing volume while box model does not. Note that in our Fig. 4b the dimension is plotted in a logarithmic scale (because the change of performance plotted against the number of dimensions is very low), which is not the case in the figure by Patel et al.

---

### Official Review · Reviewer_mG3n · 2022-07-12

**Rating:** 7
**Confidence:** 4
**Soundness:** 3 good
**Presentation:** 3 good
**Contribution:** 3 good

**Summary:**

This study explores a structured multi-label prediction problem. To this end, the authors propose to convert logical constraints into
soft geometric constraints in the hyperbolic embedding space, where the hyperplanes are viewed as convex areas, with insideness and disjointness of these regions representing logical linkages (implication and exclusion). Extensive tests on 12 multi-label classification problems demonstrate the model's capacity to boost performance.

**Questions:**

Is this method only applicable to the Poincaré disk model, and is it applicable to other hyperbolic models as well?

Is there another relationship in the data set and how to solve it?

**Limitations:**

As mentioned by the author, other logical constraints can exist in these datasets and the authors do not currently consider these relationships.

**Strengths And Weaknesses:**

Strengths: This study presents a novel translation that converts logical constraints into soft geometric constraints in the hyperbolic embedding space. Besides there is clear geometric intution where the implication is modeled by the geometric insideness while the mutual exclusion is modeled by the geometric disjointness.

Weakness:
The experiment lacks some case studies that they can faithfully show that the obtained results well reflect the initial motivation.

---

> ### Author Response · Authors · 2022-08-02
> **Response to Reviewer mG3n**
>
> **The experiment lacks some case studies that they can faithfully show that the obtained results well reflect the initial motivation.**
> We conducted an additional analysis w.r.t the correlation between the hierarchy level of labels and the magnitude of the learned label embeddings (i.e., the center points of the hyperplanes) of HMI by following the settings of MBM. The positive Spearman rank correlation scores in the table below show that HMI is able to capture the hierarchy of the labels in the taxonomy.
>
> Table 1. Spearman rank correlation between the number of descendants in the true label taxonomy with each of the following: embedding magnitude for MVM, box embedding volume for MBM and negative embedding magnitude for MHM and HMI. Baseline results are taken from [11]
> | Model | ExprFUN  | CellcycleFUN | DerisiFUN | SpoFUN | Enron   | Diatoms | Imclef07a |
> |-------|----------|--------------|-----------|--------|---------|---------|-----------|
> | MVM   | $-0.06$  | $-0.11$      | $-0.01$   | $0.06$ | $-0.11$ | $0.04$  | $-0.02$   |
> | MHM   | -0.37    | 0.38         | 0.40      | 0.38   | 0.19    | 0.31    | 0.32      |
> | MBM   | 0.47     | 0.49         | 0.50      | 0.48   | 0.47    | 0.23    | 0.43      |
> | HMI   | 0.55     | 0.49         | 0.52      | 0.49   | 0.52    | 0.43    | 0.48      |
>
>
> We also have added more comparisons with MBM. See our response to Reviewer zjFa (Q1).
>
> **Is this method only applicable to the Poincaré disk model?**
> Our idea can be extended to other hyperbolic models. Our main condition is that the hyperplane in hyperbolic spaces are "curved" (which is the case for all hyperbolic models) so that implication and exclusion can be modeled by geometric inclusion and disjointedness, respectively.
>
> **Is there another relationship in the data set and how to solve it?**
> Hierarchies and exclusions are the most ubiquitous constraints for multi-label classification. More complex constraints such as existing rules with roles $\exists C.r \sqsubset D$ can be encoded but it requires additional encoding such as embeddings of roles (or relations), which has been studied in ontology and knowledge graph embeddings, but is less often encountered in real-world settings.
> We believe this is an interesting trade-off problem and leave it for future research.

---

### Official Review · Reviewer_v6kK · 2022-07-12

**Rating:** 5
**Confidence:** 2
**Soundness:** 3 good
**Presentation:** 3 good
**Contribution:** 3 good

**Summary:**

This paper proposes to model hierarchical structure as an embedding inference using Poincare balls. Hierarchical inclusion and exclusion are used to construct training losses in Poincare space and experiments show the proposed model generally outperforms box-based baselines across multiple datasets.

**Questions:**

1. The theoretical comparison with box representation needs elaboration. For instance, intersection of boxes can represent new concepts since it remains a box shape. But intersections of Poincare plane seems not have this nice attribute. While when modeling clean structures (i.e. membership and nonmembership) Poincare has good potential, but fuzzy hierarchies seems a lack of capability.

2. This paper is not citing the NeurIPS 17 on Poincare representation of hierarchy (Nickel and Kiela). What is new/old in this paper needs to be stressed.

**Limitations:**

The amount of novelty over the NeurIPS 17 paper is unclear. This paper has a substantial amount of analysis which is a bonus to have. But the technical crux is on the model.

**Strengths And Weaknesses:**

1. The proposed Poincare hyperplane has some nice attributes on inclusion and exclusion. The construction of the training loss is intuitive and sense-making.

2. Experiments on hierarchical multilabel classification suggest promising results over box-based baselines.

---

> ### Author Response · Authors · 2022-08-02
> **Response to Reviewer v6kK**
>
> **The theoretical comparison with box representation needs elaboration. For instance, intersection of boxes can represent new concepts since it remains a box shape. But intersections of Poincare plane seems not have this nice attribute.**
> We agree that intersectional closure is a nice property in general. However, it also limits the expressiveness for embedding hierarchy and exclusion (HEX) graph. A counter example (Fig. 5 in the Appendix B) is given in the Appendix B (proof is given by [20]). The major issue of box embeddings is that the lower-way intersection may enforce higher-way intersection (e.g., pair-wise two-way intersections of three boxes enforce three-way intersection of the three boxes) because of intersectional closure,, leading to the fact that the box model is not able to represent some simple HEX graphs in any dimension. This is a phenomenon we have observed in our used dataset, e.g., in functional genomic datasets, where many proteins functions (which are labels) have pair-wise (lower-way) intersections but a group of them are not intersected.
>
> **Capability of representing fuzzy hierarchies.**
> Our model is indeed capable of dealing with fuzzy hierarchies . The defined insideness measure is sufficient to realize fuzzy concepts.
> For example,  the probability of the subConceptOf relation is represented by one ball being inside another ball to a larger or lesser extent.  We have not explored its empirical properties and in general this seems to be an area that lacks further research as well as benchmarks.
>
> **Comparison with NeurIPS 17 on Poincare representation of hierarchy (Nickel and Kiela).** Our work differs from Nickel's work with regard to both motivation and method. Nickel's work aims at improving graph embeddings but we aim at improving multi-label classification by injecting background knowledge (hierarchies and exclusions). Besides, their method only aims at embedding instances (with points) while our work targets the embedding of both instances (with points) and labels (using hyperplanes).

---

> > ### Comment · Reviewer_zjFa · 2022-08-09
> > **Theoretical Comparison with Box**
> >
> > I disagree with the characterization of the limitation of box embeddings as a "major issue".
> >
> > In particular, it seems to me the HEX graph represented in Figure 5 (Appendix B) can, actually, be represented using box models:
> >
> > https://ibb.co/6ZG2vpn
> >
> > I read the HEX graph as follows:
> > 1. Anything labeled D should be labeled A and B, and should not be labeled E or F.
> > 2. Anything labeled E should be labeled A and C, and should not be labeled D or F.
> > 3. Anything labeled F should be labeled B and C, and should not be labeled D or E.
> >
> > The box model in the link above does accurately encode these relationships. You may argue, for example, that there are regions of the label space which allow for elements to be simultaneously labeled with A, B, and C, and that is true, however that is not in contradiction with anything in the HEX graph. In a similar way, there are also parts of the space which are only labeled with A or B or C, which is also the case for the hyperbolic model. There doesn't seem to be a way to encode such a constrained with a HEX graph, one would need the notion of HEX hypergraph. Once one allows for such constraints, however, it becomes clear that hyperbolic also suffers similar limitations (eg. a constraint where items labeled with A must always be also labeled B or C, but not all three).
> >
> > Do you have empirical evidence that the "binary forcing ternary" limitation of box embeddings is a real limitation in practice?

---

> > > ### Author Response · Authors · 2022-08-09
> > > **Logical equivalence**
> > >
> > > Thanks for the comments and the example. The example is true but it lost the property of embedding logical equivalence  (which is the main strength of box model the reviewer argued), e.g., in this example you are unable to express D is equivalent to the intersection of A and B and so on. In this case, hyperbolic hyperplane can encode the same HEX graph in a similar way.
> > >
> > > We believe our new comparisons against the box model (with implication only and without any constraint) are sufficient to conclude the advantages of hyperbolic model. We believe the reason is that hyperbolic space is more suitable for embedding hierarchy and hyperplane is a linear model.

---

### Official Review · Reviewer_zjFa · 2022-07-13

**Rating:** 6
**Confidence:** 4
**Soundness:** 3 good
**Presentation:** 2 fair
**Contribution:** 3 good

**Summary:**

This paper introduces a new model for performing multi-label classification which is particularly well-suited to situations where there are elements of structure (hierarchy, mutual exclusion) in the label space. The fundamental idea is to use hyperplanes in hyperbolic space as the decision boundaries for label assignment. The main benefit of hyperbolic space over Euclidean space is that there are infinitely many hyperplanes in hyperbolic space which are non-parallel and do not intersect, thus allowing for the regions corresponding to label assignment to capture hierarchical and exclusionary patterns. The previous SOTA model used axis-aligned hyperrectangles, or "boxes", and the authors argue that such a constrained geometric region is not in alignment with the natural decision boundaries provided by neural network architectures, whereas using the hyperplanes in hyperbolic space are more amenable to the output of such encoders. A thorough analysis on 12 datasets, along with additional ablation studies are provided in support of these claims.

**Questions:**

1. On lines 39-40 the authors write that current methods ignore the importance of constraining mutual exclusion, and while they do highlight an important point (that zero implication violation could be attained by assigning the same score to all labels) it should also be noted that other models (in particular, those based on box regions) could also very easily incorporate a similar mutual exclusion loss. For example, MBM could just include a loss which measures the volume of intersection between boxes associated with mutually exclusive labels, thus encouraging these boxes to be disjoint. With this awareness, do the authors feel that it is worth noting that HMI and HMI+HLR columns in Table 1 are making use of this additional information? Alternatively, it might be relatively easy to implement and train such a method for MBM, or to rerun HMI and HMI+HLR with just the implication information to provide two additional columns in Table 1.

2. Patel et al. 2021 also included an evaluation of the models without any constraint injection, as this is the setting which may be most common in practice. Would it be possible to include such an evaluation on the 12 datasets, at least in the Appendix?

3. I did not understand the section on "Classification via hyperbolic logistic regression" (lines 188-195). Why is it desirable to consider the distances from an instance to a Poincaré hyperplane of a label as the logit? Wouldn't we want a higher score for points points which are further "inside" the n-Ball, rather than on the boundary?

### Suggestions
1. Line 114-115: The sentence here reads as though a (single) Poincaré hyperplane is the union of a hypersurface $\partial \mathbb B^n \cap \mathbb D^n$ and all linear subspaces going through the origin, which is not what is meant. Instead, consider saying that "Poincaré hyperplanes are defined by $\partial \mathbb B^n(\mathbf z) \cap \mathbb D^n$ where $\mathbb B^n(]mathbf z)$ is a ball centered at $z$ whose boundary $\partial \mathbb B^n(\mathbf z)$ intersects the Poincaré ball $\mathbb D^n$ perpendicularly. In the limit, as $|\mathbf z| \to \infty$, we obtain linear subspaces going through the origin, which are also considered that Poincaré hyperplanes, however we exclude this limiting case from consideration in practice."
2. As suggested in 1, it seems far simpler to parameterize the hyperplanes using the center of the ball rather than the "center point $\mathbf c$". I don't believe you actually need the notion of the center point, and as such you will be able to remove quite a lot of unrelated technical details (Definition 4 - which is good, because the log map hadn't been defined previously - much of the discussion in lines 119-141). Instead of the function $\mathcal C$ mapping to a center point, we can either think of it mapping to the center of the ball or simply think of it mapping to a Poincaré hyperplane itself, without specifying a particular parameterization. Many additional subscripts can be removed as well.
3. Lines 130-131: It is mentioned that Euclidean hyperplanes do not naturally allow for modeling set-theoretic semantics, but this is a bit untrue - they certainly can capture set-theoretic relationships in the same way that the hyperbolic space allows for, but they are limited in the complexity of the patterns of such relationships that they can capture. I believe the fact you wish to highlight is that the existence of infinitely many non-parallel hyperplanes in hyperbolic space which do not intersect implies that decision boundaries in hyperbolic space can capture more complicated set-theoretic interactions, such as mutual exclusivity.
4. Lines 147-148: I could not understand the sentence starting "In particular". I think what is meant is that "In particular, an $n$-ball $\mathbb B_\mathbf w$ contains $\mathbb B_\mathbf u$ if and only if $\lVert \mathbf o_\mathbf u - \mathbf o_\mathbf w \rVert + r_\mathbf u < r_\mathbf w$, and thus we can create an insideness loss by defining..."
5. Line 186: "loss of the embeddings" is not defined, perhaps what is meant is $\mathcal L_\text{inside}$ and $\mathcal L_\text{disjoint}$.

### Typos
Line 26: "works" -> "work"
Line 70: "Analog" -> "Analogous"
Line 186: "labels and if" -> "labels, if"
Line 224: "follows" -> "follow"
Line 238: "methods" -> "method"
Line 313: "constraints. We" -> "constraints we"

**Limitations:**

The authors mention that a limitation of the work is that it currently only considers implication and exclusion, and they suggest logical equivalence ($l_a \wedge l_b \iff l_c$) as another potential useful constraint to include, however I think that more can be said on this point. The fact that the current model does not include such a constraint is not merely that it has yet to be implemented, there is a representational limitation in that the intersection of two balls is not, generally speaking, another ball, and therefore the existing model does not lend itself to easily encoding this constraint. On the other hand, models which use box regions can very easily incorporate such a constraint, as the intersection of two boxes is also a box.

**Strengths And Weaknesses:**

Originality: The method is novel, and the reasons for this approach are well-motivated.

Quality: The submission is technically sound. The experimental results, while only a minor improvement quantitatively, are statistically significant. One caveat, which will be brought up in the questions, is that it is not necessarily clear that the full benefit comes from the use of hyperbolic hyperplanes, but rather the injection of the mutual exclusion constraints, which other models could also potentially easily exploit. Still, even if the model is only on-par with SOTA results when taking this into account, the underlying idea is sound.

Clarity: The submission is mostly clear, but there are a number of improvements to the presentation I would recommend in the Questions section. Even so, I feel I would be able to reproduce the model from the description provided in the paper.

Significance: The task is of interest to a number of researchers, and more broadly the idea of the incorporation of structure in Hyperbolic space would be of interest to the community of researchers working on such representations.

---

> ### Author Response · Authors · 2022-08-02
> **Response to Reviewer zjFa**
>
> **Comparision with MBM with only implication and without any constraints.**
> This is a very insightful suggestion. During the last week, we have added the comparison of two variants of our method HMI (HMI with only implication and HMI without any constraint) on the 12 datasets. Full results are shown in the updated Appendix. Our Wilcoxon test results in the table below show that HMI with only implication and HMI without any constraint still outperform their corresponding counterparts of MBM on CmAP and HCV (with p-value $<0.05$) while achieving comparable results on mAP (i.e., with better average ranks but without statistical significance, we believe this is because mAP is less sensitive to the constraints than CmAP).
>
> Table 1. Results of Wilcoxon test on HMI against MBM in the settings where only implications are available and without any constraint. $-$ means no statistical difference between the compared methods.
> | Method                              | mAP | CmAP          | HCV           |
> |-------------------------------------|-----|---------------|---------------|
> | HMI (impl.)  vs MBM (impl.)          | -   | $2.4×10^{-4}$ | $1.2×10^{-3}$ |
> | HMI (no conts.)  vs MBM (no conts.) | -   | $1.3×10^{-2}$ | $6.1×10^{-3}$ |
>
> **Why is it desirable to consider the distances from an instance to a Poincaré hyperplane of a label as the logit? Wouldn't we want a higher score for points which are further "inside" the n-Ball, rather than on the boundary?**
>
> Indeed it is as you assume: leveraging distance as logit does enforce higher score for points which are further "inside" the n-ball. This is because in (hyperbolic) logistic regression, logits are put through a sigmoid function for final classification (the classifier is $f(x) = \{l | \operatorname{\sigma}\left(h\left(x,l\right) \right) \geq 0.5, \forall l \in \mathcal{L} \}$ where $h\left(x,l\right)$ is the distance and $\operatorname{\sigma}$ is a sigmoid function). Hence, larger distance will output larger probability while $0$ distance will output $0.5$ probability.
>
> **Embedding logical equivalence.**
> We agree that intersectional closure is a nice property in general. However, it also limits the expressiveness for embedding hierarchy and exclusion (HEX) graph. A counter example (Fig. 5 in the Appendix B) is given in the Appendix B (proof is given by [20]).
> The major issue of box embeddings is that the lower-way intersection may enforce higher-way intersection (e.g., pair-wise two-way intersections of three boxes enforce three-way intersection of the three boxes) because of intersectional closure,, leading to the fact that the box model is not able to represent some simple HEX graphs in any dimension.
> This is a phenomenon we have observed in our used dataset, e.g., in functional genomic datasets, where many proteins functions (which are labels) have pair-wise (lower-way) intersections but a group of them are not intersected.
>
> **Suggestion 1-2.** We updated the definition of Poincaré hyperplane. However, we did not use the center point of n-ball as our parameterization because:
> 1) A Poincaré hyperplane can be defined by an n-ball centered at $\mathbf{z} \in \mathbb{R}^n $ only if $\mathbf{z}$ is a point outside the Poincaré ball $\mathbb{D}$  (i.e., there exists no n-ball centered inside the Poincaré ball $\mathbf{z} \in \mathbb{D}^n$ whose boundary intersects the n-ball perpendicularly).
> Hence, for optimization, we have to restrict the center points to be outside the Poincaré ball ($\mathbf{z} \in \mathbf{z} \in \mathbb{R}^n / \mathbb{D}^n$ ), which results in a tricky constrained optimization problem.
> 2) Instead, if we consider center points of the hyperplane as our parameterization, we could easily exploit Riemannian adam as our optimizer (because the center points of hyperplane are just normal points in hyperbolic space). Our proposition 1 ensures the resulting n-ball is centered outside the $\mathbb{D}^n$ while intersecting the boundary.
>
> **Suggestion 3-5 and typos.** We thank the reviewer for the detailed suggestions. We have fixed all them in the updated version.

---

> > ### Comment · Reviewer_zjFa · 2022-08-09
> > **Thank You and Follow-Up**
> >
> > **Additional experiments:** Thank you very much for running the additional experiments and reporting the results!
> >
> > **Distances:** I appreciate the clarification, however surely the distance is then a *signed* distance? My assumption is that the intention was to simply present the  model from [19], but some of the details in lines 189-196 may need to be checked more carefully.
> >
> > **Embedding Logical Equivalence:** Please see my reply [here](https://openreview.net/forum?id=XFnDhcEH9FF&noteId=-cbbD-YmdRj).
> >
> > **Center Parameterization:** This is somewhat stylistic and therefore subjective, however:
> > 1. I'm not necessarily suggesting you change the implementation, but rather the mathematical presentation. The way in which the balls are formally parameterized in the code is more of an implementation detail, and the bijection between a center point and the center of a ball can be described in the appendix, but the mathematical presentation of the model itself would, in my opinion, be much cleaner if the center was used.
> > 2. If you prefer not changing the presentation, however, then there are certain items which need to be addressed. For example, in Definition 4 you use the log map and normal vector without having introduced it previously.
> >
> > In my opinion, having a discussion right at the start of the method which goes into technical implementation details about the center point is more likely to loose interest than clarify things, and it would be better to present it in a mathematically clean way, relegating details such as how you parameterize the planes to the appendix.
> >
> > Thank you for your reply! I do not have further questions, but feel the current rating of the paper is accurate.

---

> > > ### Author Response · Authors · 2022-08-09
> > > **Center Parameterization**
> > >
> > > We thank the reviewer for the further comments. However, the main issue of using center of ball as parameterization is not w.r.t the implementation only, but rather about the presentation. Because we have to restrict the center point of ball to be located outside of the Poincare Ball, which make the presentation more complicated.
> > >
> > > We will add some explanations about the logmap and normal vectors. Thanks for the suggestions.

---

### Author Response · Authors · 2022-08-08
**Any further comments**


Dear reviewers,

Thank you again for the insightful comments and suggestions! Regarding the concerns and suggestions, we have responded to them point by point, which we believe have significantly improved the submission. In particular, we have added further comparisons against MBM in various settings (e.g., with only implication and without any constraint) and a case study showcasing that the learned embeddings exactly captured the hierarchy.
We have clarified the main differences w.r.t theoretical expressiveness of HMI and MBM, as well as some properties such as probability semantics of HMI.

We would like to ask whether the reviewers have any further comments or suggestions. We are happy to discuss further with you.

---

### Meta-Review · Area_Chair_zvWy · 2022-08-27

**Recommendation:** Accept
**Confidence:** Less certain

**Metareview:**

The reviews of this paper are uniformly positive.  The novelty is the handling of exclusion edges which expands on previous work.  On the negative side the improvements seem small and do not solidly establish the value the value of the hyperbolic hyperplanes. But the reviewers liked the paper and I recommend acceptance.



**Award:**

No

---

### Decision · Program_Chairs · 2022-09-14

Accept